# Poison frog dietary preference depends on prey type and alkaloid load

**Nora A. Moskowitz, Rachel D'Agui, Aurora Alvarez-Buylla, Katherine Fiocca**[ID]**, Lauren A. O'Connell**[ID]*****

Department of Biology, Stanford University, Stanford, CA, United States of America

* loconnel@stanford.edu

## Abstract

The ability to acquire chemical defenses through the diet has evolved across several major taxa. Chemically defended organisms may need to balance chemical defense acquisition and nutritional quality of prey items. However, these dietary preferences and potential trade-offs are rarely considered in the framework of diet-derived defenses. Poison frogs (Family Dendrobatidae) acquire defensive alkaloids from their arthropod diet of ants and mites, although their dietary preferences have never been investigated. We conducted prey preference assays with the Dyeing Poison frog (*Dendrobates tinctorius*) to test the hypothesis that alkaloid load and prey traits influence frog dietary preferences. We tested size preferences (big versus small) within each of four prey groups (ants, beetles, flies, and fly larvae) and found that frogs preferred interacting with smaller prey items of the fly and beetle groups. Frog taxonomic prey preferences were also tested as we experimentally increased their chemical defense load by feeding frogs decahydroquinoline, an alkaloid compound similar to those naturally found in their diet. Contrary to our expectations, overall preferences did not change during alkaloid consumption, as frogs across groups preferred fly larvae over other prey. Finally, we assessed the protein and lipid content of prey items and found that small ants have the highest lipid content while large fly larvae have the highest protein content. Our results suggest that consideration of toxicity and prey nutritional value are important factors in understanding the evolution of acquired chemical defenses and niche partitioning.

## Introduction

Animals must make efficient use of available food to satisfy their energetic and nutritional demands that are necessary for day-to-day function, reproduction, and survival [1]. Optimal foraging theory predicts that organisms forage to maximize fitness by reducing the energetic costs associated with consuming low-quality prey [1]. Chemically defended animals across many taxa acquire toxins through their diet, and therefore, their foraging decisions should be based on both chemical defense acquisition and nutritional quality of prey items. Most studies on organisms with acquired chemical defense focus on cataloging prey items from stomach contents, such as diet studies in bufonid (*Melanophryniscus*) toads, *Pitohui* birds and

**Data Availability Statement:** All relevant data are available on Dryad: https://doi.org/10.5061/dryad.m0cfxpp6q.

**Funding:** This work was supported by the Pew Charitable Trusts (www.pewtrusts.org; award

#00034087) and the New York Stem Cell Foundation (www.nyscf.org; award #NYSCF-R-NI58) to LAO. This work was also supported by a Student Research Grant from the Society for Animal Behavior (www.animalbehaviorsociety.org; no award number) to NAM, a postdoctoral research fellowship awarded by the National Science Foundation (www.nsf.gov; DBI-2109400) to KF, and a graduate research fellowship awarded by the National Science Foundation (www.nsf.gov; DGE-1656518) to NAM and AAB, and an HHMI Gilliam fellowship (www.hhmi.org, GT13330) awarded to AAB. LAO is a New York Stem Cell Foundation – Robertson Investigator. The funders had no role in study design, data collection and analysis, decision to publish, or preparation of the manuscript.

**Competing interests:** The authors have declared that no competing interests exist.

Natricine snakes [2–4]. However, quantification of prey availability and nutritional content are rarely considered, leaving a gap in our current understanding of how dietary preferences evolve within the context of acquired chemical defenses. Examining how prey availability and nutritional value may influence selection of prey items contributes to our general understanding of community ecology and the evolution of acquired chemical defense.

Generalist predators must strike a balance between the energetic costs of foraging and prey nutritional value and availability [5, 6]. For example, *Schizocosa* wolf spiders make choices based on their nutritional and energetic demands by selecting for certain prey, even if they are low in abundance in their environments [7]. Some predators must further consider the physiological burden of prey ingestion, especially in species that regularly consume chemically defended prey items. For example, the European Starling (*Sturnus vulgaris*) can learn to avoid toxic prey, but still choose to ingest them in the presence of undefended prey [8]. This suggests that defended prey are worth consuming despite their physiological burden, as they may be nutritionally valuable [8]. Such a nutritional tradeoff has also been proposed in possums, which regularly consume chemically defended plants that are metabolically costly but nutritionally valuable [9]. Choice of prey is arguably even more complex in organisms that acquire chemical defenses from their prey items. For example, *Acalymma vittatum* beetles have diet-derived chemical defenses and there is a tradeoff between protection from predators and the physiological costs of detoxification [6]. These choices are not only important for short term energy needs, but they can also impact development and have long lasting consequences on fitness. In *Battus philenor* butterflies, sequestering defensive chemicals from plants increases survivorship against predation as caterpillars, but at the cost of reduced fat content as adults [5]. Together, these examples suggest that environmental availability, chemical defense, and nutritional content of prey items are important considerations for predators with diet-derived defenses [2, 3, 10]. However, the interplay between these factors in foraging decisions made by animals that acquire chemical defenses from their diet is poorly understood.

Diet-derived chemical defenses have evolved multiple times in Central and South American poison frogs (Dendrobatidae), which acquire alkaloid-based chemical defenses from their diets [11, 12]. Chemical defenses have evolved at least four times within Dendrobatidae, which co-evolved with a dietary specialization on ants and mites in some species [13, 14]. Stomach content analyses have established that alkaloid-containing ants and mites constitute the majority of diet within all chemically-defended dendrobatids and some non-defended species, although there is high intra- and inter-specific variation [13, 15, 16]. Consuming an alkaloid-rich diet while also acquiring enough lipids and protein for metabolism and reproduction can be challenging. In particular, lipids are important for gamete production and metabolic maintenance in amphibians [10, 17], and yet are a limiting resource among arthropod food webs [18]. Chitinous arthropods, such as ants and mites have high levels of insoluble carbohydrates [19], and therefore frequent ingestion is expected to incur a nutritional tradeoff [20]. Although poison frog stomach content analyses have increased our understanding of diet in the dendrobatid clade, how dietary preferences based on prey phenotype or frog alkaloid load have never been tested. Thus, nutritional analyses paired with prey selectivity assays are necessary to assess the possible tradeoffs between acquiring chemical defenses and necessary nutrients.

Here, we examined prey preference of Dyeing poison frogs (*Dendrobates tinctorius*) in a laboratory setting. We tested the hypothesis that chemical defense acquisition, prey size and prey type influence dietary preferences in poison frogs. We first tested preference based on size within prey categories, as a prior study in the Malagasy Mantellidae poison frogs suggests frogs have a dietary preference towards smaller prey items [21]. Further, it is known that prey size is a crucial factor for dietary selectivity in frogs generally [22, 23]. We predicted the frogs would show a greater preference towards smaller chitinous prey categories (ants, beetles) [18,

19], but larger versions of non-chitinous prey (flies, fly larvae), as fly larvae tend to have higher protein and lipid content compared to chitinous adult arthropods [24]. We next tested preference for prey taxonomic identity and nutritional value and predicted *D. tinctorius* would prefer ants, which make up the majority of poison frog stomach contents in the wild [13, 16, 25]. Finally, we tested whether chemical defense changes prey preference by experimentally manipulating alkaloid load and measuring prey selection. We expected frogs to prefer more nutrient-rich prey as they acquire alkaloids. Together, our work helps to fill the gap in our understanding of how prey size, type and nutritional quality affect the dietary choices of chemically defended predators.

## Methods

### Animals

*Dendrobates tinctorius* (N = 20) were purchased from Josh's Frogs (Owosso, MI, USA) and housed in pairs in glass terraria with sphagnum moss, live philodendrons, shelters, egg laying sites, and a water pool. Frogs were fed three times per week with fruit flies dusted with vitamin supplements. Terraria were misted with water ten times per day and maintained in a temperature-controlled room with a 12h:12h light-dark cycle immediately preceded and followed by a 10 min period of dim lighting to simulate dawn and dusk. The Institutional Animal Care and Use Committee at Stanford University approved all frog experiments (Protocol #33839). Frogs were not euthanized for this study and were housed in our frog colony for future research.

The arthropods used in this study are comparably sized to wild prey items [26], although they are not the same species, which are not commercially available in the United States. Four groups of arthropod prey of two sizes each were used in frog feeding assays as follows (large, small): Ants: *Liometopum apiculatum* [7.0–10.0 mm, ~5 mg], *Linepithema humile* [2.2–2.6 mm, ~1.5 mg], Flies: *Drosophila hydei* [3.0 mm, ~2.5 mg], *Drosophila melanogaster* [1.6 mm, ~1.3 mg], Beetles: *Stethorus punctillum* [1.5 mm, 400–500 μg], *Dalotia coriaria* [3.0–4.0 mm, 78–80 μg], fly larvae: *Callifora vomitoria* [10 mm, ~66 mg], *Musca domestica* [3–5 mm, ~12 mg] [27, 28]. Ants were collected the morning of frog feeding trials on the Stanford University campus by sweeping ants off their nesting trees with a bristle brush into a clean plastic bag. All beetles and fly larvae were purchased from online vendors (Arbico Organics: Tucson, Arizona, USA; Evergreen Growers Supply: Clackamas, Oregon, USA; Nature's Good Guys: Ventura County, California, USA; Speedyworm Bait Suppliers, Alexandria, Minnesota, USA). Live fly larvae were kept refrigerated at 4°C and neither fed nor hydrated to prevent them from pupating into adult flies. Flies were maintained at room temperature in 32 oz. plastic deli cups with aerated lids using fruit fly media (Josh's Frogs). Beetles were stored in 32 oz. plastic deli cups with aerated lids with packaging bedding and cotton balls soaked with a 1:1 table sugar:water solution. The beetles were misted with water three times per week and occasionally were fed grain mites (*Acarus siro*) sourced from the lids of mite-infested fly cultures. To our knowledge, none of the eight prey species chosen are chemically defended. We examined chemical profiles of *L. apiculatum* and *L. humile* ants using gas chromatography / mass spectrometry (GC/MS, described below) and found no known frog alkaloids.

### Behavior

All dietary preference assays were conducted in the same room where frogs are housed, in which temperature and humidity replicate conditions of their native habitats. Before feeding trials began, all frogs were exposed to arthropod prey items six times over three weeks to reduce neophobia. Trials were conducted in a small transparent plastic bin (33 cm x 20 cm x 12 cm) stacked within an identical secondary bin with a layer of dried magnolia leaves to

simulate a forest-floor environment while keeping the texture even throughout the arena. Inner sides of the arena were coated with Fluon™ liquid teflon (Bioquip Products Inc, Rancho Dominguez, CA, USA) to prevent the arthropod prey from escaping. While piloting arena configurations, we confirmed frogs eat their regular diet of fruit flies within this setup. Of the 88 videos, 14 could not be scored due to camera malfunction. Both ingestion and an attempt to pursue or ingest prey of each type were scored using BORIS software. The scorer was not aware of the treatment group at the time of scoring.

## Size-mediated dietary preference assay

Adult frogs (11 female, 9 male; age unknown) were chosen at random and placed in the arena with 10 individual insects (five large, five small) of one of the given prey groups (ants, beetles, flies, or fly larvae). Each trial was conducted for 5 min and recorded with a GoPro camera (GoPro Inc, San Mateo, CA, USA) mounted above the arena. Each frog participated in one trial with each prey group and the order in which frogs and prey were tested was randomized. Intertrial intervals were roughly 10 min and the arena was emptied and wiped down with a moist paper towel during this time. Videos of all behavior trials were scored using BORIS (Behavior Observation Research Interaction Software) [29]. The number of "eat" events and "attempt to eat" events were scored for each size prey item within a trial.

## Alkaloid feeding, collection and taxon dietary preference assay

Dietary preference trials were conducted to determine frog preference for different insect taxa and how this preference may change with chemical defense (Fig 1A). Both male and female frogs (N = 9 male, N = 10 female) were orally administered 15 μl of a vehicle control solution (0.85% sodium chloride, 1% ethanol, 98.15% deionized water). One week later, frogs were chosen at random and placed in the arena with 20 individual insects based on which item within prey categories was most chosen by frogs in the size trials described above. The chosen candidates were: five large fly larvae (*Callifora vomitoria*), five small ants (*Linepithema humile*), five small beetles (*Dalotia coriaria*) and five small flies (*Drosophila melanogaster*). Only one prey type per category was chosen to avoid confounding taxonomic and size preferences. Each trial was conducted for 5 minutes and recorded with a GoPro camera mounted above the arena. The arena was emptied and wiped down with a moist paper towel between trials. Only female frogs (N = 10, age unknown) were used in this behavioral assay, as male frogs consistently exited the arena before trial completion. Why males exited the arena in this assay is unknown, although we hypothesize it may have been due to stress from many prey items.

To test how preferences may change as frogs acquire chemical defenses, both male and female frogs were orally administered 15 μl of 0.01% decahydroquinoline (DHQ) in vehicle solution three times over the course of one week. The following week, these "low-DHQ" female frogs were re-tested in the taxon dietary preference trials described above. One week later, both male and female frogs were orally administered 15 μl 0.01% DHQ three times per week over two consecutive weeks, representing a "high-DHQ" treatment. The following week, female frogs were retested in a final preference trial. Previous work has established this alkaloid-feeding paradigm as appropriate in creating low and high DHQ groups [30, 31]. Each female frog was assayed once for dietary preference between ants, beetles, flies and larvae at each toxicity level for a total of three trials. Although males continued to receive DHQ treatments, they were not used in behavior assays due to the consistent escape from the behavior arena during the control period.

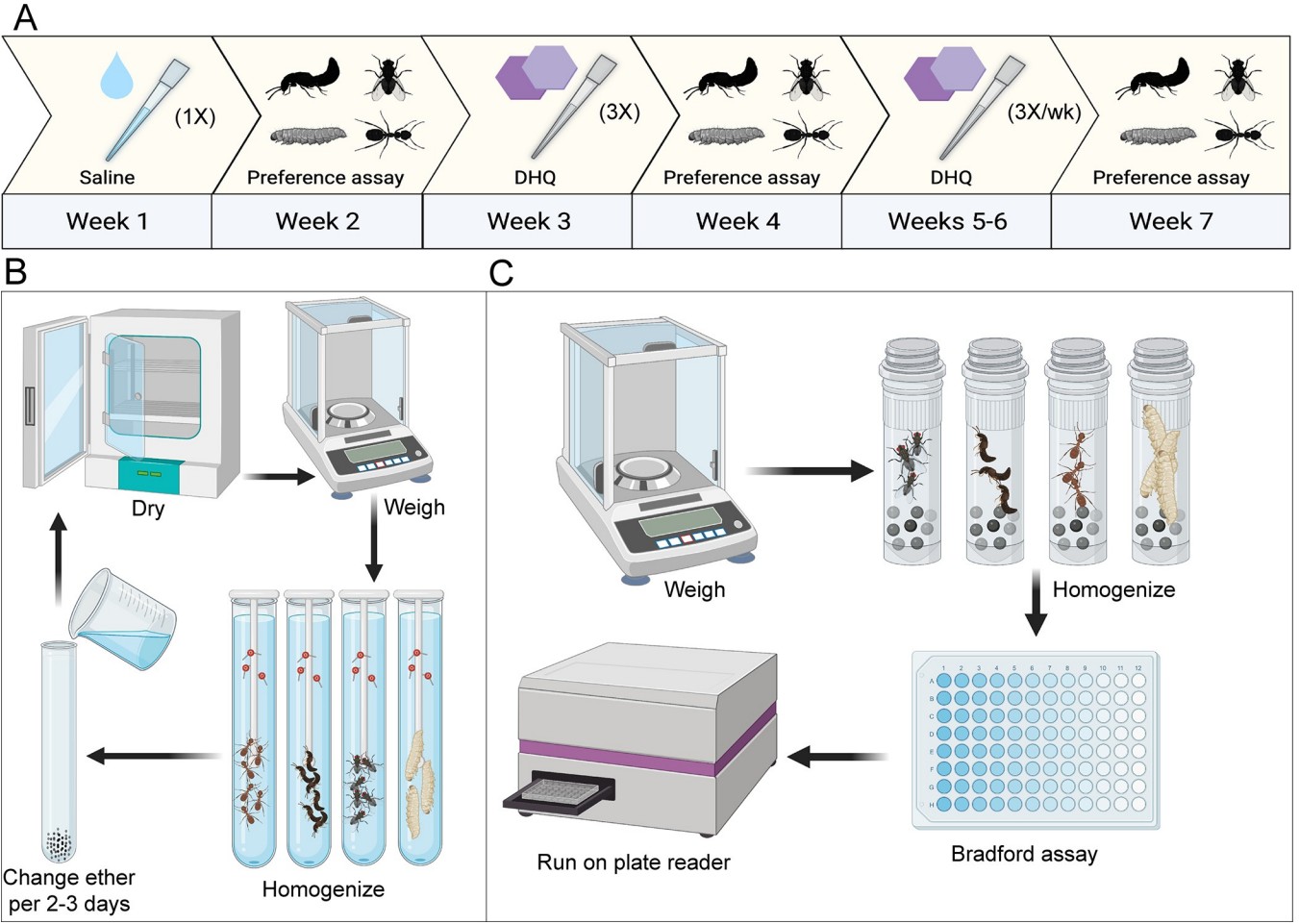

**Fig 1. Experimental workflows of diet preference and nutritional assays. (A)** The workflow of the alkaloid-mediated diet preference assay is depicted on a 7-week timeline. **(B)** Workflow of ether-soluble lipid quantification of prey items are shown. After the 6–8 day ether step, the arrow points to the drying and weighing steps, as they must be repeated to obtain lipid-free weight. **(C)** Workflow of the Bradford Protein assay of prey items are shown. Although our experiment used 5 replicates per prey group in lipid and protein assays, only single replicates are depicted here.

## Detection of alkaloids in frog secretions and ant prey using mass spectrometry

Ant alkaloids were quantified via Gas Chromatography / Mass Spectrometry (GC/MS) to determine whether ants may carry alkaloids that could influence frog dietary preference during the feeding assays. Alkaloids were extracted by soaking a volume of 1 mL of each ant species in 1 mL of methanol placed in a glass vial (Wheaton, PTFE caps, 60940A-2). After a 1-hour incubation, 1 mL of the methanol from each ant species was syringe filtered through a 0.45 um PTFE syringe filter (Thermo Scientific, 44504-NP) into a new glass vial. These methanol extractions were incubated for 24 hours at -80˚C to precipitate any lipids and proteins within the sample. A 100 uL aliquot of each sample was added to an individual GC/MS autosampler vial and the remaining samples were stored at -80˚C. GC/MS analysis was performed on a Shimadzu GCMS-QP2020 instrument with a Shimadzu 30m x 0.25 mmID SH-Rxi-5Sil MS column as described previously [16]. In brief, separation of alkaloids was achieved with helium as the carrier gas (flow rate: 1 mL/min) using a temperature program from 100 to

280˚C at a rate of 10˚C/minute. This was followed by a 2-minute hold and additional ramp to 320˚C at a rate of 10˚C/minute for column protection reasons, and no alkaloids appeared during this part of the method. Compounds were analyzed with electron impact-mass spectrometry (EI-MS), and the spectra were searched against the AMDIS (NIST) mass spectral library to identify peaks and then were visually inspected for poison frog alkaloids. All of the included compound spectra had a similarity score of above 50% in the NIST library searches.

Skin secretions from the nine male *D. tinctorius* frogs that followed an identical vehicle alkaloid-feeding regime alongside the female frogs used in the behavior trials were obtained to confirm the presence of DHQ uptake. We used males that did not participate in the taxonomic diet trials to avoid stressing the experimental female frogs, which may have influenced behavior during feeding trials. We collected alkaloids from 2–3 frogs one week after each set of feeding trials. No individual male frog had its alkaloids collected more than once. Alkaloids were collected using a small pulse stimulator (Tens®, Largo, Florida, USA) at 1 volt for 30 seconds. After stimulation, the frog's dorsal skin was wiped with a Kimwipe™ (Kimberly-Clark, Irving, Texas, USA) and then stored in 5 ml methanol in glass vials. From the methanol in which the Kimwipe™ was stored, 1 ml was syringe filtered through a 0.45 um PTFE syringe filter (Thermo Scientific, 44504-NP) into the new glass vial (Wheaton, PTFE caps, 60940A-2) supplemented with 25 ug (-)-nicotine (Sigma Aldrich, N3876). All tubes were then capped, vortexed, and stored at -80˚C to precipitate lipids and proteins. After precipitating for 24 hours, the supernatant was filtered through a 0.45 um PTFE syringe filter again into a new glass vial. We first analyzed frog secretions using GC/MS as described above and did not detect DHQ. We then used liquid chromatography / mass spectrometry (LC/MS) as previously described [30, 31] given this method is more sensitive. The preliminary LC/MS data is consistent with an increase in DHQ with longer feeding times.

## Nutritional content assays

Crude lipid content was quantified across the eight prey groups used in the dietary preference behavior experiments using an ether-soluble lipid extraction method (Fig 1B) [24]. All arthropods were fresh-frozen at -80˚C and stored for 1–2 weeks. After thawing, specimens were separated into five 1–2 mL biological replicates per prey group. The number of specimens used per prey type varied due to density and size. Each sample was placed in an incubator at 60˚C until they were dried to a constant mass, defined as less than a 0.10 percent change after an additional 20 minutes of drying [24]. One replicate per arthropod sample group was weighed once after 2 hours, and again after three hours, when a constant mass was reached for all prey groups except for fly larvae, which reached a constant mass after ~12 hours of drying. Once a constant mass was reached for all prey groups, the final mass was recorded and samples were placed in a glass culture tube filled with 5 ml of petroleum ether (Thermofisher Scientific, Waltham, MA, USA) and ground with a Dounce Homogenizer. Specimens were then incubated in ether for 8 days at room temperature. Every two days during this period, the ether was carefully decanted from each tube to prevent loss of non-lipid mass and then replaced with fresh ether. Specimens were then left in their tubes uncovered for ≥24 h in a fume hood to allow for evaporative removal of the remaining ether. Samples were then placed into an incubator at 60˚C for ≥3 h for additional drying. Once specimens were completely dried, they were weighed and lipid-free mass was subtracted from total mass, indicating the biological portion made up of ether-soluble lipids.

Crude proteins were extracted from each of the eight prey groups. All arthropods were fresh-frozen live at -80˚C and stored for 1–2 weeks. Five replicates of 10–150 mg of each arthropod type were put into 2 ml tubes, each containing 2.4 mm stainless steel beads (Omni

Life Science Inc, Raynham, Massachusetts, USA, SKU: 19–640) and 0.8 ml of cold 1x phosphate-buffered saline (PBS, pH 7.4). Samples were homogenized at 4°C using a Beadmill 24 Homogenizer (Thermofisher Scientific) with the following settings: Speed in m/s (S): 4.5, Number of cycles (C): 10, Cycle time (T): 30s, Interval paused between cycles (D): 10s. Samples were then centrifuged at 2,000 rpm at 4°C for five minutes. The supernatant containing the protein lysate was moved into a new 1.5 mL microfuge tube and stored for 1 week at -80°C. Proteins were then quantified using a Bradford Protein assay according to manufacturer's instructions (Pierce Coomassie Plus Bradford Protein Assay Reagent, Thermofisher Scientific). Samples and albumin standards were run in triplicate on a 96-well optical plate and quantified on a BioTek Synergy H1 96-well plate reader (Winooski, Vermont, USA). Absorbance was measured at 595 nm and an endpoint absorbance value was calculated for each standard and sample. Crude protein concentrations of arthropod samples were calculated based on the absorbance values of samples compared to the standard curve.

## Data analysis and visualization

Statistical analyses were run in R version 4.1.1 using the glmmTMB package [32]. To assess differences in large and small prey consumption, we fit zero-truncated generalized linear mixed models (GLMM) for each of the four prey assays (large vs. small ants, beetles, flies, fly larvae) based on a truncated Poisson distribution of the number of "eats" or "attempts to eat" across all 19 frogs. We use "attempts" because our goal is to assess preference from a time and energetic investment from the frog's perspective. Since the frogs were willing to spend time and energy pursuing prey items (often, the same individual item) over others, we felt it was important to include their attempts in our preference quantification. To control for individual variation in the subjects, frog ID was used as a random effect, and since each frog differed in its total number of prey interactions across the four size preference assay categories, "total interactions with prey" was used as an offset in order to account for these differences in consumption. Sex was also used as a covariate because of previous knowledge of differences in diet between male and female dendrobatids [33].

To assess differences of prey interactions across control, low-DHQ, and high-DHQ groups, we used our taxonomic prey preference assay data to fit a zero-truncated generalized linear mixed models (GLMM) to compare the number of prey interactions for each prey type: ants, beetles, flies, fly larvae based on a truncated Poisson distribution of the number of "eats" or "attempts" across the 10 female frogs that were assayed. To account for individual variation in the subjects, frog ID was used as a random effect. As each frog differed in its total number of prey interactions across the four prey choices, we used "total interactions with prey" as an offset in order to account for these differences in consumption. Sex was not considered as a variable in the model, since all frogs used in the taxonomic assay were female. To check for differences in the general structure of frog diet categories across the three groups, we performed a permutational multivariate analysis of variance (PERMANOVA) on Bray-Curtis dissimilarities. We used a zero-truncated GLMM (glmmTMB package, R version 4.1.1) to compare total frog interactions (eats or attempts of all prey items) across the three groups: control, low and high DHQ. The model was based on a truncated Poisson distribution as described above. To control for repeated measures in the subjects, frog ID was again used as a random effect.

P-values were adjusted using the Benjamini-Hochberg procedure to correct for multiple hypothesis testing across all dietary preference and nutritional analyses. Data were visualized with boxplots and barplots created using the package 'ggplot2' in R version 4.1.1 and infographics were created with images from BioRender.com.

## Results

### Frogs prefer smaller flies and beetles within prey groups

Overall, frogs interacted (ate or attempted) more with smaller prey items (Fig 2). Frogs interact more with small versus large flies (GLMM, Z: 16.02, BH-adjusted p <0.0001) and small versus large beetles (GLMM, Z: 2.558, BH-adjusted p = 0.021). There was no preference for small versus large ants nor fly larvae within this behavioral assay (Generalized linear mixed model (GLMM), Ants: Z: 0.844, BH-adjusted p = 0.477; fly larvae: Z: 0.712, BH-adjusted p = 0.477). The relative frequency of prey interactions did not differ between prey groups (Kruskal-Wallis chi-squared = 2.6556, df = 3, p-value = 0.448).

### Alkaloid ingestion slightly changes dietary preference

The total number of interactions across all prey types did not differ between the three alkaloid groups (PERMANOVA; F: 0.8932, p = 0.470). When frogs without DHQ were given the choice between small beetles, small flies, small ants and large fly larvae, they interacted most with large fly larvae (GLMM, Z: 4.95, p<0.00001). Additionally, high DHQ frogs interacted less with fly larvae when compared to control frogs (GLMM, Z: -2.598, BH-adjusted p = 0.036), although interactions with fly larvae were extremely variable across groups. Within all three groups, larvae were interacted with most frequently. There were no differences found among ant (GLMM, Z: 0.326, BH-adjusted p = 0.744), fly (GLMM, Z: 0.879, p = 0.379, BH-adjusted p = 0.744), or beetle interactions (z: -0.409, BH-adjusted p: 0.744) among groups.

### Prey items differ in lipid and protein content

All eight prey species were evaluated for two important dietary components: lipids and proteins (Fig 3). Overall, lipid content differed across all prey types (Kruskal-Wallis $\chi^2$ = 29.564,

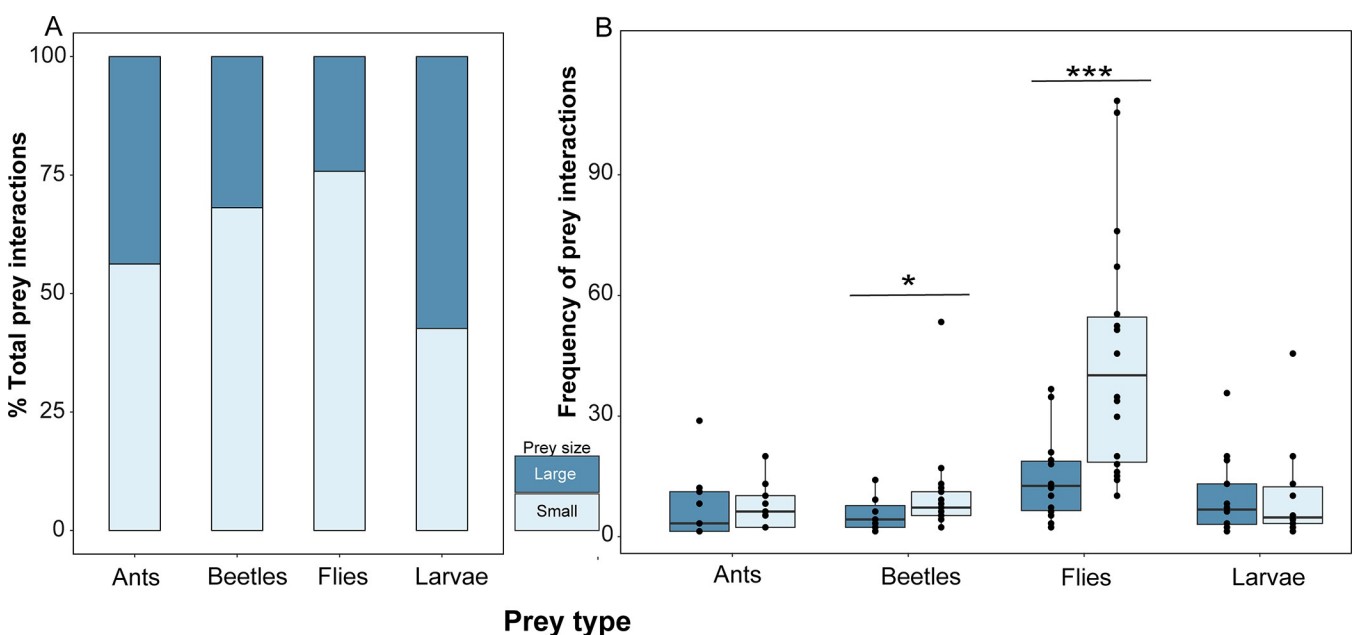

**Fig 2. Frogs preferred interacting with smaller flies and beetles.** *Dendrobates tinctorius* frogs mainly interacted with (ate or attempted) small prey items in prey size assays. Each frog was presented with one group at a time and given the choice of small and large versions of ants, beetles, flies and fly larvae. **(A)** Percent breakdown of all interactions across prey groups are plotted in a stacked bar chart and **(B)** The frequencies of individual frog interactions among large and small prey are visualized as box plots, where each dot represents a frog.

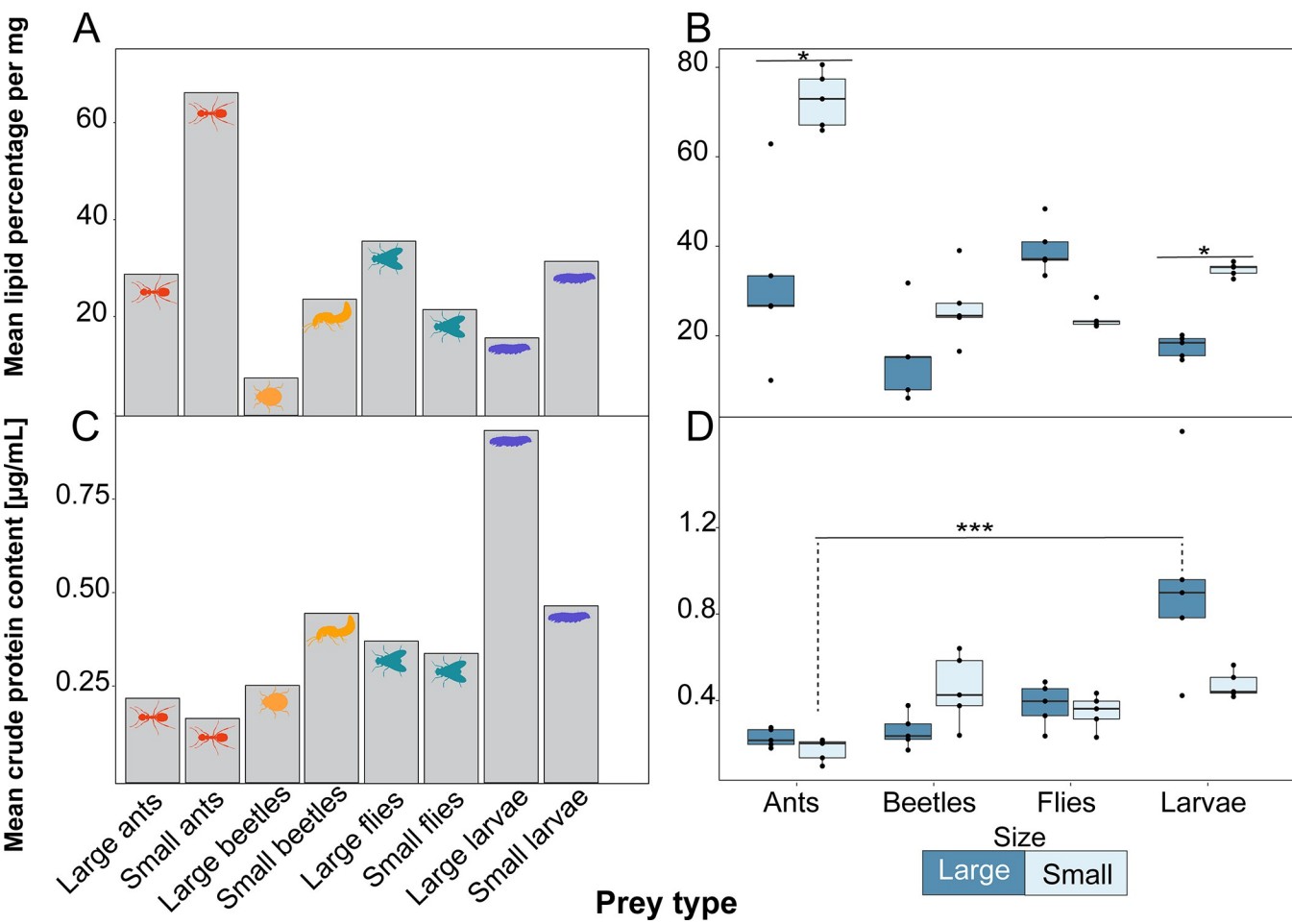

**Fig 3. Nutritional content of prey from dietary preference assays.** (A) Average crude lipid contents across replicates of the eight arthropod species are depicted in a bar chart, and (B) Variation of these values across replicates is shown in a box plot, where each dot represents a replicate. (C) Average crude protein content across replicates of the eight arthropod species are depicted in a bar chart, and (D) variation of these values across replicates is shown in a box plot, where each dot represents a replicate.

df = 7, p-value = 0.0001). Small ants (*Linepithema humile*), beetles (*Dalotia coriaria*) and fly larvae (*Musca domestica*) had a higher proportion of lipids than their larger counterparts (ant: *Liometopum apiculatum*, beetle: *Stethorus punctillum*, and fly larvae: *Callifora vomitoria*), while large flies (*Drosophila hydei*) had a higher proportion of lipids than small flies (*D. melanogaster*) (Fig 3, Table 1). Protein content also differed across species (Kruskal-Wallis $\chi^2$ = 29.206, df = 7, p-value = 0.0001), where large fly larvae have more protein than most prey items, followed by small beetles, which have more protein than small or large ants and flies (Fig 3, Table 2). Protein content did not differ within taxa for large compared to small prey. In summary, small ants have the most lipids, while large fly larvae have the most protein, of all eight species tested.

## Discussion

We tested the hypothesis that frog chemical defenses and prey traits like size and nutritional value would influence diet choices in poison frogs. We expected laboratory *D. tinctorius* to prefer small prey items, reflecting their wild diet [13–16], which was generally true. However, we

**Table 1. Differences in crude lipid content (% dry weight, mg) between arthropods.**

| Comparison | Z | Raw P | BH-adjusted P |
|---|---|---|---|
| large fly larvae—small ants | -4.0307 | 0.0001 | 0.0008 |
| large beetles—small ants | -4.1659 | 0.0000 | 0.0009 |
| small ants—small flies | 3.1109 | 0.0019 | 0.0131 |
| large flies—large fly larvae | 3.0298 | 0.0025 | 0.0137 |
| large beetles—large flies | -3.1650 | 0.0016 | 0.0145 |
| small ants—small beetles | 2.6240 | 0.0087 | 0.0348 |
| large beetles—small fly larvae | -2.6240 | 0.0087 | 0.0406 |
| large ants—small ants | -2.4617 | 0.0138 | 0.0430 |
| large fly larvae—small fly larvae | -2.4887 | 0.0128 | 0.0449 |
| large flies—small flies | 2.1100 | 0.0349 | 0.0976 |

Statistics of arthropod taxa differences in crude lipid content using Kruskal-Wallis test with a Dunn Multiple Comparison Posthoc test. P-values were adjusted using the Benjamini-Hochberg procedure to correct for multiple hypothesis testing.

were surprised to find frogs did not prefer ants over other prey items, as these frogs are thought to have a dietary specialization on ants and mites in the wild. Rather, *D. tinctorius* generally prefers prey items with high protein content, and frog alkaloid load can influence prey preferences. We discuss these laboratory results in the context of field studies in amphibians and the importance of replicating these studies under natural conditions in the future.

## Prey selectivity of arthropods

Predators' ability to consume and obtain energy from prey is primarily limited by the prey size, making size an important factor in dietary selection [1]. In our study, *D. tinctorius* frog dietary preference was assessed using feeding assays where prey varied in size. Frogs preferred small flies and beetles over large ones, with no significant preference for size in ants or fly larvae. Our findings are similar to a study conducted with *Mantella aurantiaca* poison frogs (Mantellidae), which have a slight preference for smaller prey items [21]. We did not find a size preference for ants or fly larvae, which has been documented in *Chiasmocleis mehelyi*

**Table 2. Differences in crude protein content (conc [μg/ml]) between arthropod prey.**

| Comparison | Z | Raw P | BH-adjusted P |
|---|---|---|---|
| large fly larvae—small ants | 4.1932 | 0.0000 | 0.0008 |
| large ants—large fly larvae | -3.5709 | 0.0004 | 0.0050 |
| small ants—small fly larvae | -3.4086 | 0.0007 | 0.0061 |
| large beetles—large fly larvae | -3.1246 | 0.0018 | 0.0125 |
| small ants—small beetles | -2.9217 | 0.0035 | 0.0195 |
| large ants—small fly larvae | -2.7864 | 0.0053 | 0.0249 |
| large flies—small ants | 2.5159 | 0.0119 | 0.0475 |
| large ants—small beetles | -2.2995 | 0.0215 | 0.0668 |
| large beetles—small fly larvae | -2.3401 | 0.0193 | 0.0675 |
| large fly larvae—small flies | 2.1507 | 0.0315 | 0.0882 |

Statistics of arthropod taxa differences in crude protein content using Kruskal-Wallis test with a Dunn Multiple Comparison Posthoc test. P-values were adjusted using the Benjamini-Hochberg procedure to correct for multiple hypothesis testing.

(Microhylidae) frogs that show a preference for small ants and avoidance of large ants relative to environmental abundance [23]. However, we did not test ants from the natural range of *D. tinctorius* and testing such species may reveal other preferences based on size or other prey traits like alkaloid content. Additionally, these captive frogs have been exclusively fed *Drosophila melanogaster* (small flies) until the start of this study, which may have influenced their dietary preferences. Despite these limitations, our study and previous literature show that prey size is a crucial factor for dietary choices of predators [21].

Taxonomic dietary preference was assayed using the most frequently chosen species (either large or small) from each prey group (small ants, small beetles, small flies, large fly larvae), and frogs interacted more large fly larvae over other prey items, though the difference was not statistically significant. Although poison frogs consume a diverse array of non-ant prey items [13, 14, 34], the stomach contents of *D. tinctorius* and many other chemically defended poison frogs are composed of over 50% ants [13, 26, 34]. Unexpectedly, we found that *D. tinctorius* frogs preferred fly larvae over ants, beetles, and flies. Although surprising, we must note that our lack of native prey items in these experiments could have influenced these results, as frogs may have preferences for native alkaloid-defended ants, which could not be uncovered by our study. It would be beneficial to test whether preference differs when using native prey items, as it is possible that frogs' foraging behavior would have been different when presented with ants from their natural habitat. For example, a field study with *Chiasmocleis mehelyi* (Microhylidae) frogs, which are hypothesized to sequester chemical defenses from their diet, showed preferential selection for ants over other arthropod categories [23, 35]. Frogs in general can be selective predators, as the stomach contents of bufonid toads (*Bufo cognatus* and *Bufo woodsii*) differ from local arthropod abundance [36]. However, prey availability may strongly contribute to ant abundance in poison frog diets, as frogs may occupy microhabitats of the forest with high ant abundance relative to other dietary arthropods. For example, a study that compared stomach contents of alkaloid-defended Malagasy poison frogs (Mantellidae) to environmental availability of arthropods found no significant differences between them, suggesting that their diet may reflect environmental availability rather than prey selectivity [37]. Given the known importance of prey availability, we encourage future studies on organisms with diet-derived defenses to consider comparing environmental availability of prey items' effect on dietary patterns and preferences.

### Chemical defenses and prey preference

While the acquisition of chemical defenses from diet can be physiologically burdensome, defended prey may also be nutritionally valuable. We tested the hypothesis that acquiring chemical defenses would change prey preferences and predicted that alkaloid-associated prey, such as ants, would become less desirable due to low nutritional value reported in the literature [19, 24]. In other words, we expected that overall metabolic and nutritional demand would increase with alkaloid sequestration. Contrary to our predictions, frogs decreased their preference for fly larvae, with high DHQ frogs showing a lower preference for fly larvae over control groups. It was surprising that larvae are the most frequently chosen category across groups given that the stomach contents of wild poison frogs tend to be composed of over 50% ants [14, 16, 25]. As preference changes slightly with DHQ consumption, it is possible that alkaloids can change foraging behavior as described in other animals. For example, in Red Knot birds (*Calidris canutus*), bodily levels of ingested toxins slow their feeding rates [38, 39]. Another example comes from *Mytilus* mussels, where previous ingestion of toxins slows their toxin uptake rate and feeding rate [40, 41]. Further, although not chemically defended themselves, possums consistently eat toxic, metabolically costly plants for their nutritional value [42].

Based on these studies, a positive feedback relationship could be expected between frog alkaloid levels and ingestion of alkaloid-associated prey up to a physiological threshold of chemical burden. In the future, using ant species with alkaloids that poison frogs sequesters may give more ecologically-relevant insights into how behavior changes with alkaloid uptake.

Our study has several limitations that should be accounted for in future studies. First, frog behavior may change over time with repeated exposure to prey items [43–45]. Our experimental design utilizes within-subject testing as frogs went from non-toxic to various loads of DHQ levels and thus repeated exposure and testing may reflect learning through experience rather than changes associated with alkaloid levels. We cannot verify that learning does not influence the prey preference patterns observed in our study. Second, the alkaloid load of frogs and prey items in this study do not reflect natural conditions. All eight prey species in the present study are not known to possess small-molecule alkaloids [46, 47], and we confirmed that the ants in our study do not have frog alkaloids. Moreover, while DHQ was detected in experimental frogs, it was at very low levels, being detectable by the more sensitive LC/MS but not by GC/MS. GC/MS is used by our lab and others to quantify frog alkaloids [48], suggesting that the DHQ levels in the present study are not ecologically relevant. However, it is currently unknown how alkaloid load influences frog foraging behavior. Future studies on distinct groups of chemically defended and undefended frogs being presented with native prey items that contain alkaloids the frogs sequester are needed to thoroughly understand dietary preference. Ideally, future studies on taxonomic preference would include both male and female sexes, as our taxonomic trials included females only. The results from our captive study suggest that dietary preference research across sexes would be a worthwhile line of research to pursue in the future.

## Prey selectivity and nutritional content

Dietary lipids and proteins are important for amphibian reproduction and survival. For example, the survival rate of *Epidalea calamita* toad tadpoles is positively correlated with protein consumption [49], and leopard frog tadpoles given a protein-rich diet are more likely to overcome *Batrachochytrium dendrobatidis* infection [50], a disease which has ravaged frog populations globally [51]. Lipids are also of known importance to frogs, specifically for metabolic demand and gamete production [17]. A study that monitored whole-body lipid content in domestically-reared *H. marmoratus taeniatus* frogs showed a significant decrease in whole-body lipid content following their breeding period, suggesting that fat stores are quickly depleted through reproduction [52]. Another study conducted in *Rana tigrina* found that frogs with small abdominal fat bodies showed significantly lower fecundity and decreased egg size than those with large abdominal fat bodies, and these differences were caused by diet quality [53]. Given the importance of lipids and proteins to gamete production, metabolic demand, immunity, and survival, nutrient content of prey items should be quantified when investigating factors that shape prey preference in amphibians.

In the current study, we assessed crude protein and lipid content of eight prey species, and unexpectedly found small ants were high in lipid content. Much of the literature describes ants as being composed primarily of insoluble fibers due to their chitinous exoskeletons [18, 19, 54]. On this basis, we expected eating ants to impose a trade-off in poison frog foraging decisions between acquiring alkaloids versus nutrition. Yet, there are very few studies examining ant nutritional value, although our ant lipid results are similar to one study assessing nutrient quality of small ant species, which report lipid contents of 40–60% [55]. However, our small ant lipid results are substantially higher than the only study which examined lipid percentage of the same small ant species as used in our study, *Linepithema humile*, as they report lower

lipid contents of 18–20% [56]. However, the ants in that study were given strict feeding regimens for the purposes of studying behavioral changes associated with manipulated protein : carbohydrate ratios in the diet. Therefore, those ants are expected to have a different nutritional makeup than the ones in our experiment, which are sourced from the wild. Lipid and protein content of large fly larvae (*Calliphora vomitora*) from our current study matched previously reported values [57]. Overall, the relative lipid values among orders from our study (Diptera, Coleoptera, Hymenoptera) are quite similar to the only large-scale study which quantified and compared lipid content across major arthropod orders [24]. However, a recent review of the nutritional quality of insects for animal feed shows that both lipid and protein contents are remarkably variable across studies of all insect taxa, even within single species [58]. More studies with larger sample sizes across a wider range of arthropods are necessary to understand their nutritional value as prey items. We note that a fruitful future direction would be to measure lipid, protein, and alkaloid content from prey item species within the *D. tinctorius* natural range.

Although arthropods are an important dietary component of many taxa [57, 58], there are a few studies which quantify lipid or protein content of individual arthropod species. The vast majority of studies which examine arthropod crude protein and lipid content are focused on the benefits of arthropod consumption for humans and domestic animals [59–61], with little focus on wild animal diets. Within our study, small ants and large fly larvae seem to be the most nutrient-rich of all prey items. Large fly larvae have the highest protein content but are lipid-poor, while small ants have the highest lipid content but are protein-poor (Fig 4). Both sizes of beetles and flies have similar protein levels to one another, but flies had a higher lipid content than beetles. Therefore, a nutritional trade-off arises between the decision of *D. tinctorius* individuals to consume either lipid and protein-rich prey items in the presence of both. Assuming the small ants in wild poison frog diet possess similar lipid and protein levels to the small ants in the present study (*Linepithema humilis*), chemical defense acquisition may not signify a total nutritional tradeoff, given the lipid richness of small ants. However, we encourage longer-term studies to be conducted using a higher sample size, prey items which more accurately reflect poison frog wild diet, and a wider variety of poison frog species and alkaloids, since diets between defended and undefended species vary [13].

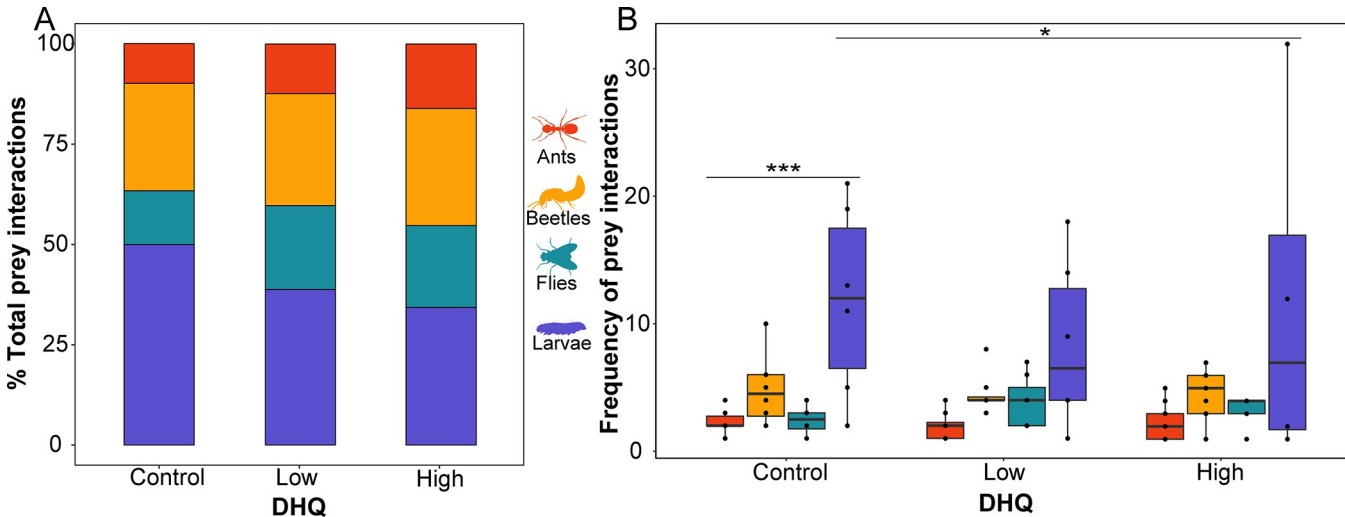

**Fig 4. Frog prey preference changes with increasing chemical defense.** (A) Percent breakdown of all prey interactions (eats or attempts) across DHQ-fed and vehicle control groups are plotted in a stacked bar chart and (B) the raw frequencies of prey interactions among frog toxicity groups are visualized as box plots to show the variation of prey interactions across individuals, where each dot represents a frog.

Prey choice becomes more complex when considering prey with high protein content, like fly larvae, in the presence of lipid-rich small ants, which frogs might innately associate with chemical defenses. Although this tradeoff has not been demonstrated, we have surmised this tradeoff from our data since we found that prey species used in this experiment differ in nutritional quality. We expected a clearer-cut tradeoff between consuming ants, which have high levels of indigestible carbohydrates [19, 57] while missing out on the nutritional value of undefended prey. However, given the lipid richness of small ants and the ant-dependent nature dendrobatid defenses, we propose that poison frogs consume alkaloid-laden prey and higher protein prey opportunistically, considering that protein is a limiting resource in arthropod food webs [18] and required for amphibian health [49, 50]. Similarly, a particularly high demand for lipids should be expected in neotropical frogs that tend to breed for the majority of the year [62, 63] and should have a high lipid demand. Poison frogs likely encounter and eat many more ants than larvae in their natural habitats, as previous literature shows >50% ants making up most poison frog stomach contents [13, 14], implying that availability may be even more important than preference in shaping poison frog diet. Since the small ants in our experiment are the most lipid-rich prey item, we propose the possibility that those in their native diets may also be lipid-rich, which could create a tradeoff for frogs given the choice between a high-lipid ant or a high-protein larvae. In our controlled lab setting, the frogs chose to interact the most with large larvae, the prey item with a substantially higher protein content than all prey categories. Taken together, these wild diet studies paired with our results suggest that typical ant-rich wild poison frog stomach contents likely reflect a combination of their ant-rich habitats and innate dietary preferences as shown in our study, and as proposed previously [13, 26, 64].

## Conclusions

We found that poison frogs prefer interacting with large larvae and that alkaloid uptake influences dietary choice, in that preference for larvae decreases with alkaloid acquisition (between control and high DHQ categories only). The known importance of lipids to amphibian reproduction and survival, taken together with our prey nutrient and preference assay results, show that poison frogs may have nutritionally benefitted from a dietary specialization on ants before they evolved an ability to acquire chemical defenses from them. We suggest that innate prey preferences, the nutritional value of prey, and prey availability are all important for our understanding of how dietary alkaloid sequestration evolved multiple times within the Dendrobatidae clade. The influence of nutritional value on dietary choices is essential for understanding the evolution of acquired chemical defenses and niche partitioning across heterotrophs. Future studies should evaluate the ecological significance of our findings by investigating preference and nutritional content of prey items available to wild frogs across a larger geographical and taxonomic scale that includes defended and undefended species.

## Supporting information

**S1 File. This excel file contains all data spreadsheets, including frog behavior, arthropod lipid and protein content, and ant alkaloid data.**
(XLSX)

## Acknowledgments

We thank all the frog caretakers that maintain our poison frog and tadpole laboratory colony and Dr. Marie-Therese Fischer for feedback on an early version of this manuscript. We thank

the Stanford University Mass Spectrometry (SUMS) core for their expert guidance and support on measuring DHQ in frogs. We acknowledge that our work takes place on the ancestral and unceded land of the Muwekma Ohlone tribe.

## Author Contributions

**Conceptualization:** Nora A. Moskowitz, Lauren A. O'Connell.

**Data curation:** Nora A. Moskowitz.

**Formal analysis:** Nora A. Moskowitz.

**Funding acquisition:** Nora A. Moskowitz, Lauren A. O'Connell.

**Investigation:** Nora A. Moskowitz, Rachel D'Agui, Aurora Alvarez-Buylla, Katherine Fiocca.

**Methodology:** Nora A. Moskowitz, Aurora Alvarez-Buylla, Lauren A. O'Connell.

**Project administration:** Lauren A. O'Connell.

**Resources:** Nora A. Moskowitz, Lauren A. O'Connell.

**Supervision:** Lauren A. O'Connell.

**Visualization:** Nora A. Moskowitz.

**Writing – original draft:** Nora A. Moskowitz.

**Writing – review & editing:** Rachel D'Agui, Aurora Alvarez-Buylla, Katherine Fiocca, Lauren A. O'Connell.

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
