## [Decision Letter · Decision Letter 0]

24 Feb 2022

PONE-D-22-02381Poison frog dietary preference depends on prey type and alkaloid loadPLOS ONE

Dear Dr. O'Connell,

Thank you for submitting your manuscript to PLOS ONE. After careful consideration, we feel that it has merit but does not fully meet PLOS ONE’s publication criteria as it currently stands. Therefore, we invite you to submit a revised version of the manuscript that addresses the points raised during the review process.

You manuscript has been reviewed by two reviewers and I have included their reports. Both liked the idea of the manuscript and that it was overall well presented. However, they also raised a number fundamental concerns, several related to methodology but also to how well your conclusions were supported by data. You need to supply convincing argument both to me and the reviewers that the problems raised do not challenge the conclusions of the study. In your response letter, make sure that you include point by point responses to the reviewers' comments.

We look forward to receiving your revised manuscript.

Kind regards,

Peter Eklöv

Academic Editor

PLOS ONE

Journal Requirements:

2. In your Methods section, please include a comment about the state of the animals following this research. Were they euthanized or housed for use in further research? If any animals were sacrificed by the authors, please include the method of euthanasia and describe any efforts that were undertaken to reduce animal suffering.

"This work was supported by the Pew Charitable Trusts (www.pewtrusts.org; award #00034087) and the New York Stem Cell Foundation (www.nyscf.org; award #NYSCF-R-NI58) to LAO. This work was also supported by a Student Research Grant from the Society for Animal Behavior (www.animalbehaviorsociety.org; no award number) and a graduate research fellowship awarded by the National Science Foundation (www.nsf.gov; DGE-1656518) to NAM. LAO is a New York Stem Cell Foundation – Robertson Investigator. The funders had no role in study design, data collection and analysis, decision to publish, or preparation of the manuscript."

"This work was supported by the Pew Charitable Trusts (www.pewtrusts.org; award #00034087) and the New York Stem Cell Foundation (www.nyscf.org; award #NYSCF-R-NI58) to LAO. This work was also supported by a Student Research Grant from the Society for Animal Behavior (www.animalbehaviorsociety.org; no award number) and a graduate research fellowship awarded by the National Science Foundation (www.nsf.gov; DGE-1656518) to NAM. LAO is a New York Stem Cell Foundation – Robertson Investigator. The funders had no role in study design, data collection and analysis, decision to publish, or preparation of the manuscript."

Reviewers' comments:

Reviewer's Responses to Questions

**Comments to the Author**

1. Is the manuscript technically sound, and do the data support the conclusions?

Reviewer #1: Partly

Reviewer #2: Partly

2. Has the statistical analysis been performed appropriately and rigorously? 

Reviewer #1: Yes

Reviewer #2: Yes

3. Have the authors made all data underlying the findings in their manuscript fully available?

Reviewer #1: Yes

Reviewer #2: Yes

4. Is the manuscript presented in an intelligible fashion and written in standard English?

Reviewer #1: Yes

Reviewer #2: Yes

5. Review Comments to the Author

Reviewer #1: The study appears to have been reasonably conducted and analyzed. I have only a few minor comments in that regard. I am, however, somewhat concerned with the interpretation of the results which I do not think are necessarily supported by the data. It is possible that I am misunderstanding the points the author is trying to make, in which case I believe that some additional clarity is warranted. Specifically, I am not certain that the claim that both toxicity and nutritional value are taken into account by frogs during prey selection is substantiated. Rather the results seem to suggest a single preference for high protein fly larvae. Specific comments related to these points follow:

Line 99-101: This sentence is confusing.

Line 102-103: This prediction isn’t backgrounded so I am unsure where it comes from.

Line 171-178: My only concern with the method is that all frogs were exposed to treatments in the same order which may be confounding. It is possible then that any differences between control, high, and low dosages are a results of experimental order. I don’t think that the control here is fully appropriate. As I understand it the control treatment was vehicle solution at the beginning of the trial to every frog. This doesn’t appear to control for time or the repeated administration of vehicle solution. I would have liked to see a control group that was given vehicle solution at the same time points as the experimental group.

Line 179: It would be helpful to know how many videos were discarded form the dataset.

Line 272-276: This section is hard to follow.

Line 312-315, 346-351: I think you need a reference here. Also I wonder if this results is due to biomass of ants being dominant in most natural settings rather than prey preference exactly. The authors do discuss this a bit later on in the discussion but I think is perhaps worth highlighting in more detail. The findings here really support the idea that prey availability not preference might be the main factor determining diet. It also occurs to me that there is an alternative hypothesis here that frogs can determine prey alkaloid content. Because all of the prey items offered here were not defended it is possible that you wouldn’t see predicted prey choice if they “know” all the options are alkaloid free.

Line 333-334: references please.

Line 360-362: It isn’t clear why this is contrary to predictions? Why would you predict that alkaloid supplemented frogs should be bias against protein rich foods?

Line 440-441: Has this been demonstrated? A naïve preference for defended prey, because as captive bred frogs that have never been exposed to nature prey sources they may not associate any species with being defended unless it was innate.

Line 449-452: I am not convinced your data supports this conclusion. It seems like there is no evidence from your experiments that dietary preference explains diet of wild populations.

Line 455-459: Again if I understand your results right, I don’t think this is supported. The preference toward larvae suggests a preference for high protein not high lipid prey items doesn’t it?

General: I would specify fly larvae throughout as all the prey species exhibit a larval stage.

Fig 2b: I am having a hard time interpreting this figure. The caption doesn’t seem sufficient to explain what is being show. Does this not look like a preference for adult flies and not larva?

Reviewer #2: The authors designed several experiments to better understand foraging decisions by a frog that acquires its chemical defenses from diet. Overall, I think the ideas presented in the paper are significant, and represent a new and important direction towards an understanding of organisms that acquire chemical defenses. The literature review presented in the paper, and the synthesis of the experimental results as they pertain to current knowledge in the study system are well presented in the discussion. My main concern with the paper has to do with the biological relevance of some of the results, and what they might mean in a natural system. I also have several questions surrounding some of the methodology, all of which are aimed at improving the overall study. I present my concerns below. To the authors credit, they do address in the discussion some of the limitations of their lab experiments as they pertain to natural systems, which I think is important. Although just an opinion, I think the most significant results in the manuscript are in relation to the protein and lipid content analysis of different prey items. These results are novel (at least within this system), and are more widely applicable to natural systems.

Specific Comments:

Arthropod prey used in experiment.

The authors used two different ant species in their feeding experiments, Camponotus vicinus and Linepithema humile, which are assumed (to the authors knowledge) not to be chemically defended. It is possible that these species do not contain alkaloids, however the present authors recently report in Moskowitz et al. (2020) the presence of alkaloids in other species of ants in the same genera. Although there are many exceptions, the presence of alkaloids in ants are commonly found to be genus specific. Therefore, it seems possible that the ants used in the present study also contained alkaloids. To eliminate this possibility, the two ant species used in the present experiment should be examined for alkaloids, as the presence of alkaloids would significantly impact the findings of the study.

Size mediated dietary preference assay.

Why did the authors use both “eat” and “attempt to eat” events? If the goal of the study was to understand what the frogs would eat when given a choice, then why use “attempt to eat”? I am sure there is a good reason, but in the absence of more information on the category “attempt to eat” (e.g., what constitutes an attempt to eat? how was this determined?), it is not clear. Also, what are the raw numerical differences between these two categories? Did most frogs eat, and only a small subset attempt to eat? Or, did most frogs attempt to eat? Given the aim is dietary preference, it seems that the eat category would be the only relevant category to measure. A failed attempt to eat could be interpreted as a mistake by a frog, but it could also be a decision by the frog not to consume a particular prey item. Again, without more information, this is difficult to assess.

Alkaloid feeding and taxon dietary preference assay.

Can the authors offer any ideas on why all of the males in this experiment exited the arena before trial completion? This result is especially strange given that 9 males did not exit the arena in the size-mediated experiments. Also, the fact that this part of the experiment only involved females should be added to the discussion.

Lines 103-105: (1) “non-chinous” should be “non-chitonous.” (2) As written, the prediction is a bit confusing and it is not entirely clear what the authors mean.

Lines 128- 133: A minor point, but there is no information on where the fruit flies originated.

Lines 143-144: Something is missing or incorrect. How is it possible that “one week before the trials began”, frogs were “exposed to arthropod prey items six times over three weeks”?

Lines 143-149: Were the frogs exposed to the novel arthropod prey items in their larger terraria, or were they first put in experimental arenas? Also, is it know if frogs will eat “normally” after being removed from a larger terrarium and placed into a smaller arena? Was a control performed, in which frogs were simply fed their normal diet of fruit flies in the experimental arenas? I think this would strengthen the findings.

Line 152: I think “five big” should read “five large”.

Lines 167-168: Why was a limited prey selection chosen for the alkaloid feeding experiment, as compared to the size-mediated experiment? Adding some clarification would strengthen this portion of the experiment.

Lines 171-178: Did the authors examine the frogs for alkaloid content at the end of the experiment? Although there was a small effect of alkaloid “load” on prey interaction, the differences in alkaloid amount between frogs is not reported. My concern is that there was not a difference in alkaloids between treatments, or that the differences were too small to be biologically relevant. The authors cite Alvarez-Buylla et al., which conducted a similar type of alkaloid feeding experiment over the course of 5 days. However, in the present paper, the “low-DHQ” treatment was fed to frogs 3 times, whereas the “high-DHQ” treatment was fed to frogs 6 times. The difference between treatments is only 3 feedings (that occurred over a longer time period for the high treatment), which may not have been large enough to result in a difference in alkaloid content between frog treatments. Comparing the alkaloid content in frogs between treatments would provide more clarification.

Line 179-180: (1) “Videos” should be “videos” (2) How many videos could not be scored due to camera malfunction? The number or percentage should be reported. Also, how does this change the overall sample size of the experiment?

Line 254: (1) Why are the results heading different than the method headings? (2) The heading is confusing, as both small and large things are prey. But, more importantly, the result stating that there “was no preference for small versus large ants or larvae within this behavioral assay” contradicts the heading.

6. PLOS authors have the option to publish the peer review history of their article (what does this mean?). If published, this will include your full peer review and any attached files.

Reviewer #1: No

Reviewer #2: No

---

## [Author Response · Author response to Decision Letter 0]

3 Aug 2022

Response to Editor and Reviewer Comments

We have addressed and/or incorporated all reviewers’ comments in our revised manuscript. The major concern was interpretation of our data regarding preference for large fly larvae in our taxonomic preference trials. We have clarified this in our responses and in our revised manuscript.

We have also added two additional co-authors, Aurora Alvarez-Bullya and Katherine Fiocca, who performed experiments to address reviewer concerns. These new experiments include the extraction and quantification of alkaloids from ant prey and frogs.

Reviewer #1: The study appears to have been reasonably conducted and analyzed. I have only a few minor comments in that regard. I am, however, somewhat concerned with the interpretation of the results which I do not think are necessarily supported by the data. It is possible that I am misunderstanding the points the author is trying to make, in which case I believe that some additional clarity is warranted. Specifically, I am not certain that the claim that both toxicity and nutritional value are taken into account by frogs during prey selection is substantiated. Rather the results seem to suggest a single preference for high protein fly larvae. Specific comments related to these points follow:

Line 99-101: This sentence is confusing.

Response: Sentence has been reworded and split for clarity.

Line 102-103: This prediction isn’t backgrounded so I am unsure where it comes from.

Response: Thank you. This had been mentioned prior in the manuscript (references 18-19, 24) (lines 91-92). However, we have fixed this sentence for clarity and added citations: “We predicted the frogs would show a greater preference towards smaller chitinous prey categories (ants, beetles) (18-19), but larger versions of non-chitinous prey (flies, larvae), as insect larvae tend to have higher protein and lipid content compared to chitinous adult arthropods (24).”

Line 171-178: My only concern with the method is that all frogs were exposed to treatments in the same order which may be confounding. It is possible then that any differences between control, high, and low dosages are a results of experimental order. I don’t think that the control here is fully appropriate. As I understand it the control treatment was vehicle solution at the beginning of the trial to every frog. This doesn’t appear to control for time or the repeated administration of vehicle solution. I would have liked to see a control group that was given vehicle solution at the same time points as the experimental group.

Response: Thank you for your input. We agree that a non-toxic control group, in addition to our within-subject control approach, would have been ideal. Unfortunately, this experiment was conducted during the COVID-19 pandemic when frogs were not available for purchase. Therefore, our approach was the optimal way to test preference at a variety of alkaloid ingestion levels with limited animals. We discuss the limitations of this approach at length in the discussion.

Line 179: It would be helpful to know how many videos were discarded from the dataset.

Response:

Of 88 videos, 14 could not be scored. This has been added to the manuscript.

Line 272-276: This section is hard to follow.

Response: We have rewritten the paragraph with more context and explanation surrounding our statistical reports. To summarize, control frogs preferred interacting with fly larvae more than high DHQ frogs. Across all three groups, the total number of all prey interactions did not differ. We acknowledge that our heading was misleading, and we have adjusted it accordingly.

Line 312-315, 346-351: I think you need a reference here. Also I wonder if this results is due to biomass of ants being dominant in most natural settings rather than prey preference exactly. The authors do discuss this a bit later on in the discussion but I think is perhaps worth highlighting in more detail. The findings here really support the idea that prey availability not preference might be the main factor determining diet. It also occurs to me that there is an alternative hypothesis here that frogs can determine prey alkaloid content. Because all of the prey items offered here were not defended it is possible that you wouldn’t see predicted prey choice if they “know” all the options are alkaloid free.

Response: Thank you, references have been added. We appreciate your thoughtful interpretation of the data, and agree that prey availability is likely a dominating factor in what poison frogs consume, and that there is a likely abundance of ants in comparison to other arthropod prey. We have added another sentence to the discussion which highlights this, as well as your point that true preference may not have been adequately tested given that alkaloid-laden prey items were not presented to the frogs in feeding trials:

“It is possible that frogs’ foraging behavior would have been different when presented with ants from their natural habitat, and we acknowledge that our experimental design does not allow us to assess true preference, which must be tested using native prey items. “

Line 333-334: references please.

Response: Reference has now been added.

Line 360-362: It isn’t clear why this is contrary to predictions? Why would you predict that alkaloid supplemented frogs should be bias against protein rich foods?

Response: Thank you. This is because in the wild, poison frog diet usually consists of >50% ants. This has been clarified in the text with citations added.

Line 440-441: Has this been demonstrated? A naïve preference for defended prey, because as captive bred frogs that have never been exposed to nature prey sources they may not associate any species with being defended unless it was innate.

Response: No, this has not been demonstrated, rather it reflects what we have surmised from our data since we demonstrated that prey categories differ in nutritional quality. Therefore, we have edited this paragraph to reflect the speculatory nature of the statement: “Prey choice becomes more complex when considering highly proteinous prey, like fly larvae, in the presence of lipid-rich small ants, which frogs might innately associate with chemical defenses. Although this tradeoff has not been demonstrated, we have surmised this tradeoff from our data since we found that prey species used in this experiment differ in nutritional quality. "

Line 449-452: I am not convinced your data supports this conclusion. It seems like there is no evidence from your experiments that dietary preference explains diet of wild populations.

Response: We did not intend to imply that our results directly explain diet in native populations. Rather, we suggest that what we learned from innate behavior in a controlled, laboratory setting should be considered in the context of poison frog prey consumption and in tandem with our previous knowledge. Although ants make up the majority of poison frog diet, our results suggest that they consume protein-rich prey opportunistically, which implies that ants are likely more abundant/available when compared to a protein-rich prey item, such as fly larvae.

Line 455-459: Again if I understand your results right, I don’t think this is supported. The preference toward larvae suggests a preference for high protein not high lipid prey items doesn’t it?

Response: Your interpretation of the results is correct. We intended to write this as speculation regarding how poison frogs would behave in their natural habitats in the presence of both prey items, and have reworded it to clarify that this is merely speculatory. Although it is known that frogs consistently eat small ants and fly larvae, we include this speculation because we feel it is important in the context of what we know about poison frogs in their natural habitat, and that this new insight may be informative. We know that poison frogs likely encounter (and therefore eat) many more ants than larvae in their natural habitats, as previous literature shows >50% ants making up most poison frog stomach contents. Since the small ants in our experiment are the most lipid-rich prey item, we propose the possibility that those in their native diets may also be lipid-rich. Yet, in our controlled lab setting, they choose to interact the most with large larvae, the prey item with a substantially higher protein content than all prey categories. 

Reviewer #2: The authors designed several experiments to better understand foraging decisions by a frog that acquires its chemical defenses from diet. Overall, I think the ideas presented in the paper are significant, and represent a new and important direction towards an understanding of organisms that acquire chemical defenses. The literature review presented in the paper, and the synthesis of the experimental results as they pertain to current knowledge in the study system are well presented in the discussion. My main concern with the paper has to do with the biological relevance of some of the results, and what they might mean in a natural system. I also have several questions surrounding some of the methodology, all of which are aimed at improving the overall study. I present my concerns below. To the authors credit, they do address in the discussion some of the limitations of their lab experiments as they pertain to natural systems, which I think is important. Although just an opinion, I think the most significant results in the manuscript are in relation to the protein and lipid content analysis of different prey items. These results are novel (at least within this system), and are more widely applicable to natural systems.

General: I would specify fly larvae throughout as all the prey species exhibit a larval stage.

Response: “Larvae” has now been changed to “fly larvae” throughout the manuscript.

Fig 2b: I am having a hard time interpreting this figure. The caption doesn’t seem sufficient to explain what is being show. Does this not look like a preference for adult flies and not larva?

Response: Thank you, that is correct. We agree that the caption was not sufficient and have expanded it for clarity.

Arthropod prey used in experiment.

The authors used two different ant species in their feeding experiments, Camponotus vicinus and Linepithema humile, which are assumed (to the authors knowledge) not to be chemically defended. It is possible that these species do not contain alkaloids, however the present authors recently report in Moskowitz et al. (2020) the presence of alkaloids in other species of ants in the same genera. Although there are many exceptions, the presence of alkaloids in ants are commonly found to be genus specific. Therefore, it seems possible that the ants used in the present study also contained alkaloids. To eliminate this possibility, the two ant species used in the present experiment should be examined for alkaloids, as the presence of alkaloids would significantly impact the findings of the study.

Response: Thank you for this suggestion. First, we would like to note that we misidentified our large ants in our initial manuscript submission, which had been identified as Camponotus vicinus, but are in fact Liometopum occidentale (Velvety tree ants). To address these concerns, we examined the chemical profiles of both the small and large ants and have confirmed that they are free of frog alkaloids. This is now noted in the manuscript. The raw data files and spreadsheet will be available on DataDryad and included upon acceptance. 

Size mediated dietary preference assay.

Why did the authors use both “eat” and “attempt to eat” events? If the goal of the study was to understand what the frogs would eat when given a choice, then why use “attempt to eat”? I am sure there is a good reason, but in the absence of more information on the category “attempt to eat” (e.g., what constitutes an attempt to eat? how was this determined?), it is not clear. Also, what are the raw numerical differences between these two categories? Did most frogs eat, and only a small subset attempt to eat? Or, did most frogs attempt to eat? Given the aim is dietary preference, it seems that the eat category would be the only relevant category to measure. A failed attempt to eat could be interpreted as a mistake by a frog, but it could also be a decision by the frog not to consume a particular prey item. Again, without more information, this is difficult to assess.

Response: Thank you for noting this. We use attempts because we are looking at preference from a time and energetic investment from the frog’s perspective. Since our frogs are willing to spend time and energy pursuing prey items (often, the same individual item) over others, we felt it was important to include their attempts in our preference quantification, because attempts exemplify preferential interaction with the prey. In size trials, 943 of 1733 behavior events (eats or attempts) were “attempts to eat” (54%). In taxonomic preference trials, 189 of 409 behavior events (eats or attempts) were “attempts to eat” (46%). We have added an explanation regarding why we used “attempts” in the manuscript. The raw numbers of attempts and eats across all trials are available on our supplementary spreadsheet.

Alkaloid feeding and taxon dietary preference assay.

Can the authors offer any ideas on why all of the males in this experiment exited the arena before trial completion? This result is especially strange given that 9 males did not exit the arena in the size-mediated experiments. Also, the fact that this part of the experiment only involved females should be added to the discussion.

Response: The size feeding trials include a total of 10 insects in the arena per trial, while the taxonomic feeding trials include a total of 20 insects in the arena per trial. Given that there are double the amount of insects (which was necessary, in order to adequately test preference within the four prey groups in taxonomic trials), frogs in taxonomic trials experienced much more sensory stimuli in taxonomic trials. We hypothesize that the male frogs were quick and consistent in their reaction to leave the arena, due to these stimuli. However, we cannot confirm this and are unsure of why this happened. The following sentence was added to the methods: “Why males exited the arena in this assay is unknown, although we hypothesize it may have been due to stress from many prey items.”

Lines 103-105: (1) “non-chinous” should be “non-chitinous.” (2) As written, the prediction is a bit confusing and it is not entirely clear what the authors mean.

Response: (1) Fixed. (2) A clarifying statement has been added to supplement our predictions: “We predicted the frogs would show a greater preference towards smaller chitinous prey categories (ants, beetles) (18-19), but larger versions of non-chitinous prey (flies, fly larvae), as fly larvae tend to have higher protein and lipid content compared to chitinous adult arthropods (24). We next tested preference for prey taxonomic identity and nutritional value and predicted D. tinctorius would prefer ants, which make up the majority of poison frog stomach contents in the wild (13,16,25). “

Lines 128-133: A minor point, but there is no information on where the fruit flies originated.

Response: Line 139, we state that the fruit flies originate from the company, “Josh’s Frogs.”

Lines 143-144: Something is missing or incorrect. How is it possible that “one week before the trials began”, frogs were “exposed to arthropod prey items six times over three weeks”?

Response: Thank you for noting this mistake. The “one week” portion of text was incorrect, and has been removed.

Lines 143-149: Were the frogs exposed to the novel arthropod prey items in their larger terraria, or were they first put in experimental arenas? Also, is it known if frogs will eat “normally” after being removed from a larger terrarium and placed into a smaller arena? Was a control performed, in which frogs were simply fed their normal diet of fruit flies in the experimental arenas? I think this would strengthen the findings.

Response: Thank you. The frogs were introduced to novel arthropod prey in their larger terraria, not the smaller arenas. During our pilot phase of testing various arena set-ups, frogs were tested to make sure they ate their normal diet of fruit flies within the arena. A sentence stating this has been added to the manuscript. 

Line 152: I think “five big” should read “five large”.

Response: Thank you, this has been fixed.

Lines 167-168: Why was a limited prey selection chosen for the alkaloid feeding experiment, as compared to the size-mediated experiment? Adding some clarification would strengthen this portion of the experiment.

Response: Thank you. This was done to avoid confounding taxonomic and size preferences, which we believed was necessary to test separately. We narrowed down the categories as follows. From each prey category in the size-mediated preference assays, we chose one species to represent each of the four categories. The species chosen per category was whichever one was chosen most by the frogs during the size mediated assays. This has been clarified in the manuscript.

Lines 171-178: Did the authors examine the frogs for alkaloid content at the end of the experiment? Although there was a small effect of alkaloid “load” on prey interaction, the differences in alkaloid amount between frogs is not reported. My concern is that there was not a difference in alkaloids between treatments, or that the differences were too small to be biologically relevant. The authors cite Alvarez-Buylla et al., which conducted a similar type of alkaloid feeding experiment over the course of 5 days. However, in the present paper, the “low-DHQ” treatment was fed to frogs 3 times, whereas the “high-DHQ” treatment was fed to frogs 6 times. The difference between treatments is only 3 feedings (that occurred over a longer time period for the high treatment), which may not have been large enough to result in a difference in alkaloid content between frog treatments. Comparing the alkaloid content in frogs between treatments would provide more clarification.

Response: Thank you for this comment. We have the following points in reply:

1. (Minor) First, we would like to point out that the difference between low and high DHQ groups was six feedings, not three feedings. 

2. (Major) We have now analyzed the DHQ content of the frogs used in this study. Through the experiment, skin alkaloids were quantified from male frogs that followed an identical vehicle control + alkaloid-feeding regime alongside the female frogs that were used in the trials. These frogs did not participate in the final diet trials, and their alkaloid collection happened at the end of the experiment using an electric pulse stimulator at 1 volt for 30 seconds. We first analyzed these samples using GC/MS, which is the method we use for quantifying alkaloids in frogs from the wild. However, DHQ was undetectable with that method, which highlights the reviewer’s concern that DHQ levels were very low.

3. (Major) Since DHQ was undetectable using GC/MS, we next used LC/MS methods for DHQ detection given this approach is more sensitive. We confirmed DHQ presence in frogs using this method. The raw data files generated from LC/MS machinery will be available on DataDryad upon submission. 

4. (Major) We acknowledge that behavioral differences may have been more apparent if alkaloid load differed more substantially between groups. However, we have limited knowledge of how alkaloid load influences foraging behavior, and we have added this as a suggestion for future research in the discussion. 

Line 179-180: (1) “Videos” should be “videos” (2) How many videos could not be scored due to camera malfunction? The number or percentage should be reported. Also, how does this change the overall sample size of the experiment?

Response: Thank you for noting this. Of 88 videos, 14 could not be scored, and this has been added to the manuscript. Sample size does not change substantially, mainly because most malfunctions happened during the size trials (which includes all frogs, male and female), in which each individual frog was tested 4 times – once per prey category. 

Line 254: (1) Why are the results heading different than the method headings? (2) The heading is confusing, as both small and large things are prey. But, more importantly, the result stating that there “was no preference for small versus large ants or larvae within this behavioral assay” contradicts the heading.

Response: (1) Thank you, the heading format has been fixed. (2) Thank you for noting this inconsistency. The heading has been changed to reflect accuracy.

---

## [Decision Letter · Decision Letter 1]

5 Oct 2022

Poison frog dietary preference depends on prey type and alkaloid load

PONE-D-22-02381R1

Dear Dr. O'Connell,

We’re pleased to inform you that your manuscript has been judged scientifically suitable for publication and will be formally accepted for publication once it meets all outstanding technical requirements.

Kind regards,

Mainul Haque

Academic Editor

PLOS ONE

Additional Editor Comments (optional):

Reviewers' comments:

Reviewer's Responses to Questions

**Comments to the Author**

1. If the authors have adequately addressed your comments raised in a previous round of review and you feel that this manuscript is now acceptable for publication, you may indicate that here to bypass the “Comments to the Author” section, enter your conflict of interest statement in the “Confidential to Editor” section, and submit your "Accept" recommendation.

Reviewer #1: All comments have been addressed

Reviewer #3: All comments have been addressed

2. Is the manuscript technically sound, and do the data support the conclusions?

Reviewer #1: Yes

Reviewer #3: Yes

3. Has the statistical analysis been performed appropriately and rigorously? 

Reviewer #1: Yes

Reviewer #3: Yes

4. Have the authors made all data underlying the findings in their manuscript fully available?

Reviewer #1: Yes

Reviewer #3: Yes

5. Is the manuscript presented in an intelligible fashion and written in standard English?

Reviewer #1: Yes

Reviewer #3: Yes

6. Review Comments to the Author

Reviewer #1: (No Response)

Reviewer #3: (No Response)

7. PLOS authors have the option to publish the peer review history of their article (what does this mean?). If published, this will include your full peer review and any attached files.

Reviewer #1: No

Reviewer #3: **Yes: **PRASHANTA KUMAR MANDAL

---

## [Editor Report · Acceptance letter]

7 Nov 2022

PONE-D-22-02381R1 

Poison frog dietary preference depends on prey type and alkaloid load 

Dear Dr. O'Connell:

I'm pleased to inform you that your manuscript has been deemed suitable for publication in PLOS ONE. Congratulations! Your manuscript is now with our production department. 

Kind regards, 

on behalf of

Dr. Mainul Haque 

Academic Editor

PLOS ONE